# Barriers to Accessing Eye Health Services in Suburban Communities in Nampula, Mozambique

**DOI:** 10.3390/ijerph19073916

**Published:** 2022-03-25

**Authors:** Dulnério B. Sengo, Neves A. Marraca, Alcino M. Muaprato, Sofía García-Sanjuan, Pablo Caballero, Inmaculada López-Izquierdo

**Affiliations:** 1Faculty of Health Sciences, Lúrio University, Nampula City 3100, Mozambique; dulnerio@yahoo.com.br (D.B.S.); marracanevesalbino@gmail.com (N.A.M.); alcinomuaprato87@gmail.com (A.M.M.); 2Department of Community Nursing, Preventive Medicine and Public Health and History of Science, University of Alicante, 03690 Alicante, Spain; 3Department of Nursing, University of Alicante, 03690 Alicante, Spain; sofia.garcia@ua.es; 4Department of Physics of Condensed Matter, Optics Area, University of Seville, 41012 Seville, Spain; ilopez9@us.es

**Keywords:** eye health, barriers to access, eye health services, Mozambique

## Abstract

Globally, an estimated 2.2 billion people are visually impaired (VI) or blind, and a large proportion (90%) of those affected live in low- and middle-income countries (LMICs), where access to eye health services is limited. This study aimed to identify barriers to accessing eye health services and associated factors in suburban communities of Nampula. A cross-sectional community-based study was carried out on adults ≥18 years old. A total of 338 adults were randomly selected from three communities (Muthita, Piloto, and Nthotta). Individual interviews were carried out and socio-demographic data, eye symptoms, date of last eye examination, and barriers to access to eye health services were extracted. Among participants, 49.4% had eye symptoms and 41.7% did not have their eye examinations up to date. The most cited barriers were crowding in hospitals (40.7%), financial difficulties (30.0%), self-medication (20.5%), traditional treatment (17.8%), and buying eyeglasses on the street (11.6%). Barriers limited the service target to 33%. Lower levels of schooling and monthly family income and farmer occupation were statistically associated with the most barriers as risk factors. The use of eye health services was lower due to barriers to accessing eye services. More specific intervention plans and greater cooperation between sectors are needed to improve these indicators.

## 1. Introduction

Globally, an estimated 2.2 billion people are visually impaired or blind, of which at least 1 billion have a visual impairment (VI) that could have been prevented or has yet to be addressed. A large proportion (90%) of affected people live in low- and middle-income countries (LMICs) where access to eye health services is limited [1,2,3].

These people live with reduced vision because they do not receive specialized care for the eye problems that afflict them, such as refractive errors (RE) and cataracts, which, in turn, are the biggest causes of visual impairment in the world [1,4]. Visual impairment affects considerably the individual’s quality of life, interfering with daily activities such as walking, cooking, eating, bathing, and recognizing faces. Adults with visual impairment often have lower rates of participation and productivity in the workplace and higher rates of depression and anxiety [1,5].

Furthermore, currently, many people, families, and communities continue to suffer the consequences of limited access to quality and affordable eye care services, thus having to live with visual impairment or blindness [3].

The appropriate use of eye health services is essential to reduce the burden of visual impairment worldwide [6]. It requires routine eye examinations at the recommended regularity, and is generally influenced by psychological, socio-cultural, and economic factors [6,7].

One of the aims of the World Health Organization’s (WHO) Global Action Plan 2014–2019 for Universal Eye Health was to generate evidence regarding the provision of eye health services that serve to design plans and policies to strengthen universal access to eye health [8].

Recent research in low- and middle-income countries (LMICs) has identified costs, a perceived lack of need, lack of information about the location of services, transportation difficulties, and fear of adverse outcomes as factors hindering access to eye health services [9].

The “one student, one family” program (1E1F) is a Lúrio University program offered by the Faculty of Health Sciences as a mandatory subject called “Family and Community Health”, a pioneering and innovative community outreach program, which aims to pursue an educational experience oriented towards community practice by providing critical training and guidance for future generations of health workers in Mozambique [10]. During the program, each student is responsible for a family for which they must monitor their health status, looking at the different socio-cultural and economic aspects that are determinants for the health and well-being of the respective family, performing prevention and health promotion actions at the individual and family level, and developing socio-anthropological research that will serve to guide intervention actions in communities [11].

To date, there is only one study, published in 2015, on the barriers to accessing eye health services in Mozambique (with a more quantitative aspect), and none provides a qualitative approach to the facts [12]. However, it is known that qualitative research allows greater involvement of patients and, with that, a better and more realistic approach, as it brings to light aspects associated with health services and the patient’s own experience [13,14].

Therefore, this study aimed to identify barriers to accessing eye health services and associated factors in adults belonging to communities covered by the 1E1F program in Nampula.

## 2. Materials and Methods

Nampula province is the most populous province in Mozambique, with 20.6% (5,758,920 inhabitants) of the general population, located in the northeast of Mozambique. Nampula City is the most populous district in the province of Nampula (with 760,214 inhabitants) and is also the provincial capital [15].

The province of Nampula is basically rural, where farming is the the main occupation (68.3%), followed by small seller (8.6%), non-farm worker (8.2%), and farm worker (5.0%) [16].

Lúrio University (LU) is located to the west on the outskirts of Nampula City. The program designed by LU serves the communities of Mutauhanha, which is a suburb of Nampula City, located in the Administrative Post of Muatala. This neighborhood is mainly composed of non-farm workers and popular middle classes. Annually, one or several neighboring communities are randomly selected until the capacity of the 1E1F program is covered. During 2019, the selected communities were Muthita, Piloto, and Nthotta, with a total of 2750 adults and 1290 families. Therefore, the study population was composed of adult individuals covered by the 1E1F program. For an estimation of an unknown proportion, as there is no data for Mozambique, a 95% confidence and a 5% precision were considered for a minimum sample size of 338 participants. Randomly, participants were contacted until the required sample was obtained. Inclusion criteria were age ≥18 years, covered by the 1E1F program, and available to travel to the university clinic for an eye exam and sign an informed consent to participate in the study.

### 2.1. Ethical Aspects

This study was approved by the Institutional Committee of Bioethics for Health of Lúrio University (CIBSUL), with ref: 29/Oct/CBISUL/19, in October 2019. All participants were previously informed about the nature of the study and participated in the study by signing an informed consent form. This study followed the principles of the Declaration of Helsinki.

### 2.2. Data Collection

Data collection was conducted from November 2019 to February 2020, at the University Clinic of Visual Health at Lúrio University.

### 2.3. Interview

#### 2.3.1. The Preparation of the Interview

Prior to the interview, a rapid review in PubMed and Google Scholar about the possible barriers to the access or use of eye health services was performed; the keywords defined as barriers to accessing eye health services were extracted, both for the eye examination as well as treatment (optical, drug, or surgical), which served as a guide for the interviews. The most common barriers found in the literature were cost, transportation, lack of information or education in eye care, fear of treatment, traditional treatment, trust, lack of time, not being aware of the problem, need for escort, and unhappy with medical services [6,17,18,19,20,21,22,23]. Nevertheless, a qualitative study with data saturation was carried out to collect the existence of more barriers. In the scope of qualitative research, data saturation is understood as the point at which a certain diversity of ideas is collected and at each additional interview or observation no other elements appear; but, while new data or ideas continue to appear, the search does not stop [24]. Therefore, this guarantees that the researcher has conducted the research to the point of obtaining new and important knowledge about the studied phenomenon [25]. Therefore, during qualitative research, the sample size is not chosen at the beginning, but only when the research is completed, as this sampling is not oriented towards the number of participants, but rather towards the wealth of data provided by the participants [26,27]. These interviews were transcribed word for word to obtain all possible barriers.

#### 2.3.2. Performance

After signing the informed consent, during the anamnesis, individual interviews were carried out, simultaneously with the clinical eye examination at LU. First, the socio-demographic data of each patient were collected and, subsequently, they were asked if they had any ocular symptoms or if they had their eye examination up to date (as recommended by the American Optometric Association and American Academy of Ophthalmology, which is at least 2 years for patients without risk factors and aged 18–64 years and annually for patients with age ≥ 65 years or with risk factors) [7,28]. In the qualitative study, if participants had symptoms, they were asked why they had not sought medical help earlier. If eye examinations were not up to date, they were asked if they knew the recommended frequency of examinations and the reasons behind why they did not have the examinations up to date. Barriers were identified during the first interviews using the qualitative saturation method. In the quantitative study, the interview procedure was similar; however, the interviewer completed the form and collected the frequency with which barriers were mentioned but always in an open interview.

The team of interviewers consisted of three properly trained optometrists (D.B.S., N.A.M., and A.M.M.).

### 2.4. Data Analysis

The interviews were in Portuguese, and those corresponding to the qualitative study were recorded and transcribed by the researchers D.B.S. and A.M.N until data saturation was reached, that is, when the new interviews no longer added new information regarding the barriers [25]. However, the analysis was carried out individually and, subsequently, jointly by D.B.S., N.A.M., and A.M.M. in order to ensure better triangulation between authors and to obtain all possible barriers.

A socio-demographic description of the sample was made, and the frequency of symptomatic cases, eye examinations being up to date, and barriers to accessing eye health services were determined.

The association between dependent (barriers to access) and independent variables (gender, age, education level, address, family income) was quantitatively studied through the odds ratio (OR) and adjusted odds ratio (aOR) for all independent variables; 95% confidence intervals were calculated. All data analyses were performed using SPSS version 23.0 (SPSS Inc., Chicago, IL, USA).

## 3. Results

The study included 338 individuals residing in the communities of Muthita, Piloto, and Nthotta, and only 2% of the people contacted were not present at the University Clinic on the appointed day (seven people).

Most participants were from Piloto (43.2%), aged between 18 and 81 years (with a mean of 41.37, SD 16.48), and males predominated with 54.1% of participation. The predominant school level was secondary (39.3%) and primary (28.4%), and there was a greater proportion of participants with a monthly family income of MZN <5000 (36.7%) (MZV 5000 is about USD 78.32 in 2022). As for occupation, there was a greater proportion of manual workers (20.7%) (Table 1).

### 3.1. Eye Symptoms and Eye Examination up to Date

Among participants, approximately half had eye symptoms. The ORs showed a statistical association with the age group >65 years (OR:4.6) and the occupations of farmer (OR:2.1) and retired (OR:5.7) as risk factors. However, in the multivariate model, the aOR showed as risk factors family income of MZN <5000 (aOR:3.6) and home in Piloto (ORa:2.0), while manual (aOR:0.5) and domestic worker (aOR:0.3) were protective (Table 1).

Regarding having an eye examination up to date, 41.7% of the participants did not have their eye examinations up to date; the OR showed a statistical association with manager (OR:0.1) and retired (OR:0.3) occupations as protective factors, while manual (OR:1.9) and domestic (OR:2.6) work, a low level of education, and low family income were risk factors, with the lower the level of education or family income, the greater the risk of a lack of eye examination being up to date. However, in the multivariate model, the aOR showed the age group >65 years (aOR:0.4) as being a protective factor, while low family incomes were risk factors, with a lower income representing a greater risk of a lack of eye examination being up to date.

### 3.2. Barriers

#### 3.2.1. Qualitative Analysis

It took 25 interviews to reach data saturation, that is, five interviews that did not add new data. The barriers mentioned by the interviewees were crowding in hospitals, financial difficulties, self-medication, traditional treatment, buying eyeglasses on the street, fear of treatment, waiting for disease to advance, did not think it was necessary, distrust of professionals, lack of time, transportation, thought there was no solution, lack of knowledge, and lack of someone to accompany them.

Through the interviews, we could observe that the participants were quite annoyed with the crowding in the hospitals and having to endure long queues to access eye health care.

E1: *“It takes a lot of persistence to get care in public hospitals, you have to arrive very early and queue up, if you don’t have patience, your disease gets worse right there waiting”*.

E8: *“If you have a medical appointment, you have to book every day because you won’t be able to do anything else, you arrive at the hospital in the morning and only leave there in the afternoon, and I need to work to get something for my children to eat”*.

Financial difficulty was reported as a limitation to accessing private eye health services as well as purchasing eyeglasses.

E3: *“The doctor prescribed me eyeglasses and I went to buy the eyeglasses at the optician and found them at very high prices... between seeing poorly and dying of hunger I prefer to see poorly”*.

Some interviewees opted for self-medication to bypass the long queues at public hospitals and the lack of time.

E5: *“I prefer to go directly to the pharmacy, it’s less stressful for me, I get there and explain what I feel and they give me the medicine right away, it doesn’t take me even 15 min... to face that queue at the hospital only when I feel it’s something very serious”*.

In some cases, the preference for traditional medicine was a family practice that was passed on from generation to generation.

E19: *“My grandparents died when they were over 70 years old and they never lost their sight or had surgery, they used medicine, herbs and plants, and I also grew up like that and I have no problems”*.

The prices of eyeglasses in the formal market (clinics and opticians) were not accessible to everyone, which was why some found a solution in the informal market and bought eyeglasses on the street for a low price and fast acquisition.

E7: *“I end up buying eyeglasses on the street for lack of choice, the price is more affordable, they are two or three times cheaper”*.

E9: *“I bought my last eyeglasses on the street because I was afflicted and on the street everything is fast and flexible, just arrive and try on the eyeglasses until you find one that provides better vision, pay and leave”*.

The fear of hospital treatment came from reports of unsuccessful surgeries and unsatisfactory results within communities, especially with regard to cataract surgical interventions, which caused some to seek other options.

E14: *“I don’t see myself undergoing surgery, I’m afraid, I’ve been applying a medicine (homemade herb extract) that has helped me to relieve the symptoms and I feel good”*.

E1: *“There are a lot of people there in the neighborhood who had surgery and got worse, they can’t see anything anymore. I can still see something, I prefer to continue like this as long as possible”*.

Some interviewees believed that there was a conflict of interest regarding optical prescriptions (eyeglasses) in medical appointments being very biased since the sale of eyeglasses has been a business of the eye specialists themselves, producing a distrust in professionals.

E13: *“Sometimes, in medical appointment, it does not matter what symptom the person has, at the end of the appointment they always find a way to convince you to buy eyeglasses. It gives me the feeling that it’s a business for them so they try to sell eyeglasses at all costs”*.

It was common for patients already diagnosed with cataract or pterygium not to undergo surgery, claiming to wait for the disease to advance, contrary to medical recommendations.

E12: *“They say I have to undergo surgery, but I can still see, go to the farm, do my thing, so I prefer to wait for it to mature, it’s not quite advanced yet”*.

Some did not think it was necessary to have an eye examination, as they did not have any eye symptoms.

E13: *“I really don’t feel the need to go to the doctor, put up with long queue while I see well and feel nothing”*.

The lack of knowledge about the recommended frequency for an eye examination, combined with the absence of symptoms, was cited by some as a justification for not accessing eye health services.

E15: *“I don’t know how long I should go to the doctor for a routine exam, so I only go when I feel something in my eyes, but as I see well, I am never worried”*.

Some interviewees expressed a certain conformism with their eye condition and thought that there was no longer a solution to their eye problems, which discouraged them from accessing eye health services.

E14: *“Nothing can be done in these eyes, I’ve tried everything and age doesn’t help either, each appointment I go the situation only gets worse, I’m used to it, I’m just getting by anyway”*.

Some did not have anyone to accompany them to the hospital, which constituted a barrier not only for locomotion but also for scheduling the appointment and getting directions.

E19: *“On the road there is a lot of agitation, at my age and situation it is not easy, leaving the house to go up to the hospital alone. My son and grandson usually accompany me or make an appointment for me, but often they do not have time”*.

Transport limitations arose from the financial difficulty in bearing the cost of transport to the hospital.

E12: *“The hospital is far away, to get there I need to take two buses and spend twenty (20) meticais round trip, I prefer to use this money for food at home”*.

#### 3.2.2. Quantitative Analysis

The frequency with which barriers were mentioned is shown in Figure 1. Barriers were classified according to Tanahashi’s model into four categories: availability, accessibility, acceptability, and contact [29]. The crowding in hospitals and financial difficulties were the two main barriers to accessing eye health services, mentioned by 40.5% and 29.9% of respondents, respectively. Self-medication, traditional treatment, and buying eyeglasses on the street occupied the next three positions; below 10% were the remaining barriers (fear of treatment, waiting for disease to advance, did not think it was necessary, distrust of professionals, lack of time, transportation, thought there was no solution, lack of knowledge, and lack of someone to assist).

Table 2 and Table 3 show barriers to accessing eye health services and associated factors. Overall, 67.5% of respondents mentioned some barrier to accessing eye health services. The ORs showed a statistical association with illiterate (OR:3.1), primary (OR:3.5) and secondary (OR:3.0) schooling levels and farmer occupation (OR:8.3) as risk factors, while the 45–65-year age group (OR:0.3) and teacher (OR:0.4) or manager (OR:0.3) occupations were protective factors. However, in the multivariate model, only illiterate (aOR:5.4) and primary (aOR:3.2) schooling levels and monthly income of MZN <5000 (aOR:5.1) were risk factors.

The crowding in hospitals had a statistical association with the age groups of 18–44 years (OR:11.5) and 45–65 years (OR:9.2); illiterate (OR:4.17), primary (OR:5.1), and secondary (OR:3.5) school levels; family incomes of MZN <5000 (OR:13.2), MZN 5000–14,000 (OR:12.6), and MZN 15,000–24,000 (OR:5.5); and manual work (OR:3.2) as risk factors, while being a manager (OR:0.1) was a protective factor. However, in the multivariate model, only the age groups of 18–44 years (aOR:12.9) and 45–65 years (aOR:11.0), family incomes of MZN <5000 (aOR:12.9) and MZN 5000–14,000 (aOR:8.7), and manual work (aOR:2.2) were risk factors.

Financial difficulty was statistically associated with illiterate (OR:8.8) and primary (OR:6.3) school levels, family incomes of MZN <5000 (OR:7.3) and MZN 5000–14,000 (OR:3.4), and occupation as a farmer (OR:3.8) as risk factors, while being a student (OR:0.4) was protective. However, in the multivariate model, only family incomes of MZN <5000 (aOR:12.5) and MZN 5000–14,000 (aOR:6.2) and being a teacher (aOR:3.0) were risk factors.

Self-medication was statistically associated with the age groups of 18–44 years (OR:6.1) and 45–65 years (OR:5.4), primary (OR:4.2) and secondary (OR:2.9) school levels, and family income of MZN 5000–14,000 (OR:11.1) as risk factors. However, in the multivariate model, the age groups of 18–44 years (aOR:22.2) and 45–65 years (aOR:19.6) and family incomes of MZN <5000 (aOR:9.2) and MZN 5000–14,000 (aOR:12.3) were factors of risk.

Traditional treatment had a statistical association with female gender (OR:1.8), illiterate (OR:22.4) and primary (OR:6.6) school levels, and family income of MZN <5000 (OR:6.1) as risk factors, while the age groups of 18–44 years (OR:0.1) and 45–65 years (OR:0.2) were protective factors. However, in the multivariate model, only family income of MZN <5000 (aOR:10.5) was statistically associated as a risk factor.

Buying eyeglasses on the street was statistically associated with illiterate (OR:7.4) and primary (OR:2.9) school levels, family incomes of MZN <5000 (OR:9.4) and 5000–14,000 (OR:5.8), and being a farmer (OR:6.2) as risk factors, while the age groups of 18–44 years (OR:0.1) and 45–65 years (OR:0.4) were protective factors. However, in the multivariate model, family incomes of MZN <5000 (aOR:15.1) and 5000–14,000 (aOR:11.9) and being a teacher (aOR:7.3) were risk factors, while the age group of 18–44 years (aOR:0.1) was protective.

Fear of treatment was statistically associated with the illiterate (OR:24.9) school level, being a farmer (OR:9.5), and retired (OR:5.5) as risk factors, while the age groups of 18–44 years (OR:0.0) and 45–65 years (OR:0.1) and manual work (OR:0.1) were protective factors. In the multivariate model, the age groups of 18–44 years (aOR:0.1) and 45–65 years (aOR:0.1) were protective factors, while farmer occupation (aOR:4.9) represented a risk.

Additionally, the other barriers (lack of time, distrust in professionals, waiting for disease to advance, did not think it was necessary, lack of knowledge, thought there was no solution, lack of someone to accompany, and transport) were statistically associated with illiterate (OR:3.1), primary (OR:3.5), and secondary (OR:3.0) school levels and being a seller (OR:2.2) as risk factors. However, in the multivariate model, only being a seller (aOR:2.2) represented a risk factor for other barriers.

## 4. Discussion

This study provides information regarding the proportion of eye symptoms, eye examinations being up to date, and barriers to accessing eye health services. The frequency of eye symptoms is an important indicator of the population’s eye health status, although it is subjective. Eye examinations being up to date demonstrates the utilization level of eye health services. Access barriers represent what prevents them from using eye health services with the recommended frequency and seeking solutions for their eye problems.

The assessment of eye health services was part of the World Health Organization (WHO) Global Action Plan 2014–2019 for Universal Eye Health and is critical to monitoring progress, identifying priorities, and advocating greater political and financial commitment from stakeholders and member states with regard to eye health [8].

In the present study of 338 participants, 141 (41.7%) did not have their eye examination up to date, taking into account the recommendations of the American Optometric Association and American Academy of Ophthalmology (periodicity of at least 2 years for patients without risk factors and aged between 18–64 years and annually for patients aged ≥65 years or with risk factors) [28,30]. In a study carried out in Ghana [20], similar results were found, although 3 years were used as a parameter: 40.0% of the participants did not have their eye examination up to date at the time of the research. In studies carried out in Southwest Nigeria [19] and Ethiopia [6], the percentages were higher (81% and 76.2%, respectively) compared to our study. However, the Nigeria study was carried out in rural areas, where access conditions were obviously more difficult, while in Ethiopia, Hawassa City contains a very limited number of specialized hospitals with eye health services (only one specialized hospital and four private clinics) compared to Nampula City (in turn, with seven hospital units and six private clinics), which implies a greater limitation of access to services.

The lack of eye examinations being up to date reflects the lack of access to eye health services by the communities involved.

### 4.1. Eye Symptoms

On the other hand, the age group of >65 years, family income of MZN <5000, residence in Piloto, and occupations of farmer and retired were risk factors for the presence of ocular symptoms, while manual and domestic work were protective. However, with advancing age (>65 years) and retirement, there is a greater demand for eye health services due to a greater occurrence of degenerative diseases and, consequently, eye symptoms in this group.

### 4.2. Eye Examination up to Date

Therefore, the lack of eye examinations being up to date was statistically associated with the age group >65 years and occupations of manager and retired as protective factors, while low levels of schooling and monthly family income and manual and domestic work were presented as risk factors for the lack of eye examinations being up to date. These results corroborate with those found in Ethiopia [6], in which advanced age, awareness of the importance of eye examination, and higher monthly family income statistically offered greater probability of using eye health services.

Therefore, people with a higher level of education are probably more aware of the importance of the eye examination and have better economic conditions, while people aged >65 years and retired have more free time and enjoy priority care in public services, thus constituting a predisposition to the adequate use of eye health services, thus having eye examinations being up to date.

On the other hand, manual and domestic work were protective factors for the occurrence of ocular symptoms and a risk for the lack of eye examinations being up to date, showing a certain tendency for asymptomatic individuals who do not seek eye health services, while low income was presented as a risk factor both for the lack of eye examinations being up to date as well as for the presence of symptoms; therefore, this group, even having symptoms, does not seek eye health services.

### 4.3. Barriers

The assiduity with which people seek eye health services is often associated with the barriers they have to face to access them. People from different places face different barriers to accessing eye health services. Therefore, the type of barrier depends on who they are, where they are, and the cause of the eye problem [17].

In our study, the barriers to accessing eye health services most cited by respondents were crowding in hospitals (40.7%), financial difficulties (30.0%), self-medication (20.5%), traditional treatment (17.8%), and buying eyeglasses on the street (11.6%).

In a study carried out in Nampula [22], published in 2015 and focused on refraction services and purchase of eyeglasses, patients with visual impairment were interviewed, and financial difficulty was identified as the main barrier to the use of refraction services (53%), followed by a lack of need felt (20%), distance/transport (15%), and a lack of awareness (13%). In that study, financial difficulties had a higher proportion compared to our study, which can be associated with the characteristics of the sample, since the socioeconomic conditions of the sample were, however, worse than in our study. Of the participants, 75.9% had a family income of USD <2 dollars per day (equivalent to MZN 4000 per month). A considerable proportion of participants resided in rural areas (32.8%); during the data collection period (between 2012–2013) transport conditions in Nampula were worse than in 2019–2020 (the period of realization of our study), which may justify a higher proportion of “distance/transportation” in relation to our study (3.9%). Added to this, 88.3% of participants in 2012–2013 did not even have a secondary level of education, which may be allied to a higher proportion of “lack of awareness”. The lack of felt need was higher in 2012–2013, which may be associated with occupation, since most participants (45%) in the 2012–2013 study were farmers and the visual demand in this activity is relatively lower. However, the comparison between our study and that published in 2015 has certain limitations: The latter only interviewed visually impaired people and was focused on refractive problems while our study addressed eye health as a whole and not having visual impairment was not an exclusion criterion to participate in the interview.

Similar results were found in Nigeria (in three rural communities of Edo State), in which the main identified barriers to the use of services were a lack of need felt (33.3%) and financial difficulty (26.7%), which may also be associated with the fact that the study took place in rural communities and the majority of participants were farmers, which implies a lower visual demand [31].

Financial difficulty was among the two main barriers in the three studies. Therefore, in Nampula, where eye examination and surgery in public hospitals are free, only the purchase of eyeglasses and eye drops implies costs for the patient.

However, Mozambique is on the list of countries with a low Human Development Index (0.456), occupying the 181st position (out of 189 states) in the global ranking [32]; according to the Global Multidimensional Poverty Index (MPI), 49.9% of the Mozambican population lives in severe poverty, with just over 60% living on less than USD $1.90 a day and just over 80% of the population living on less than USD $3.10 a day. Nampula province is one of those with the highest proportion of people living in severe poverty, with 55.5% [33]. Therefore, most people in Nampula do not have the economic conditions to access private services, as it implies higher costs, which cause a huge demand in the public sector, which results in overcrowding in hospitals. However, in search of a solution to their health problems, many opt for more affordable alternatives such as self-medication, traditional treatment, or buying low-quality eyeglasses in the informal market.

The purchase of eyeglasses on the street, without a doctor’s prescription, was not only motivated by overcrowding in hospitals, but also by the more affordable prices and flexibility in acquisition. The search for traditional treatment is also allied to strong historical–cultural links, supported by the beliefs of local populations, in addition to the difficulty of accessing conventional health services [34]. Traditional medicines for treating eye diseases identified in some studies have been biological derivatives, usually of animal or human origin (such as breast milk, saliva, urine, or cod liver oil), plants (herb extracts, palm wine, palm oil, bitter cola extract), and chemical substances (holy water, anointed oil, black stone, salt solution, sugar solution, antimony, and kerosene) [35,36]. The indiscriminate use of these substances in developing countries may be responsible for the increased occurrence of infections and corneal ulcerations in these regions and, consequently, an increase in cases of visual impairment [35].

In a study carried out in the Ntotta community in 2017 (Nampula) on self-medication in adults and associated factors, 54.3% of the participants self-medicated and were associated with lack of information and poor care in public hospitals as aggravating factors [37]. In our study, in addition to a lack of time, self-medication was also associated with hospital care (overcrowding in public hospitals). Therefore, self-medication is recognized as a public health problem worldwide due to the negative impact on treatment, delaying the institution of a really effective therapy, and risk of intoxication, in addition to contributing to microbial resistance and increasing the risk of eye infections [35,37].

One of the least mentioned barriers by the interviewees in this study was the lack of information and transport difficulties, which can be justified by the study location (peripheral neighborhood of Nampula City), where transport conditions are relatively better compared to rural areas, given that, according to Massarongo-Jona [38], in Mozambique access to health services is highly conditioned by the long distances to be traveled to the health unit, and even traveling these distances, there are no guarantees of care with the required quality and human dignity, given the overcrowding and conditions of care in hospitals. In terms of access to information, community health education is one of the pillars of the 1E1F program, through which health care information is disseminated, including the recommended frequency of eye examinations, diseases, symptoms, risk factors, and promotion of eye health services in communities. Therefore, this information can make a difference between adults covered and not covered by the 1E1F program. However, the data collected on barriers and up-to-date reviews were not affected as the program works in each edition with different communities.

In recent decades, the availability of eye health services has increased significantly worldwide due to the efforts of various non-governmental agencies and the national blindness prevention program. Despite this, the lack of accessibility remains a concern, especially among disadvantaged groups [39]. The barriers identified in this study are a resistance to the eradication of preventable or treatable visual impairment. However, it became evident that some barriers result from the country’s socio-economic situation; overcoming them depends on the country’s own development dynamics and requires a multisectoral effort and effective collaboration to strengthen eye health in communities.

There is still a weakness of the health system in providing quality services to the community. The private sector has supplied part of this deficit, but the use of private services entails costs and, in this scenario, the socioeconomic condition of the individual plays a crucial role in their eye health. Therefore, it became evident that communities with greater social deprivation and low income perceive more barriers and have reduced access to eye health services, thus being more vulnerable to the occurrence of ocular morbidities and vision loss.

#### 4.3.1. Barriers to Access to Eye Health Services for the Elderly

Age played an important role in the occurrence of certain barriers to accessing eye health services. In the multivariate analysis, age >65 years (elderly) was defined as a reference factor. However, the other age groups (18–44 years and 45–65 years) were protective factors for traditional treatment, buying eyeglasses on the street, and fear of treatment with respect to the elderly. Therefore, the elderly are more likely to opt for traditional treatment, buy eyeglasses on the street, and be afraid of hospital treatment (especially surgical interventions). In the elderly, there is a higher prevalence of cataract and pterygium, whose treatment is essentially surgical. Reports of unsuccessful surgery cases cause a certain fear of treatment within the community; consequently, causing evasion of conventional eye health services and the search for alternative services. Added to this, elderly people tend to be more conservative with respect to their habits and customs, with more rooted, historical–cultural ties; for this reason they seek more traditional treatment than younger people.

#### 4.3.2. Health Service Coverage and Its Evaluation

The study of barriers allows us, using the Tanahashi model [39,40], to assess equity of access and barriers to achieving universal health coverage (UHC) with equity. Figure 2 shows the operational curve. The result shows that barriers associated with availability and accessibility decrease the potential coverage by 57%. Barriers related to acceptability and contact accelerate this decrease to 66%. Tanahashi’s analysis adds the percentage of effective treatments. This calculation of effective coverage was not our objective. However, with the barriers alone, coverage would decrease to below 33% in this peri-urban population. These coverages are likely to be lower in rural areas of Mozambique.

However, this study has limitations: The sample of the present study was not representative of the population of Nampula province, since it involved communities residing in peripheral neighborhoods of the City of Nampula where the socio-economic conditions are relatively better compared to the general situation of Nampula province where most of the inhabitants live in rural areas and have agriculture as their livelihoods. However, the participants of this study were less disadvantaged compared to the general population, which supposes that many of our results could be underestimated with respect to the situation of the whole province, that is, for example, the barriers of access to eye health services are possibly worse in rural areas.

The results of the present study could serve as indicators to monitor the eye health status of the community, and future studies will measure the impact of the program at the eye health level. Intervention studies aimed at improving access to eye health services in underserved communities would be equally important. Studies measuring effective coverage would be needed to determine, together with the results of this study, the true extent of health coverage in eye health, as well as clinical epidemiological studies on the eye health status of communities, especially in the Piloto community, to find out what makes this community more prone to eye symptoms.

Research on attitudes and practices regarding eye care would be opportune to better frame educational actions that aim to make people aware of symptoms, eye diseases, and healthy practices for the eyes, as well as periodic eye examinations to improve access to eye health services and ensure early detection of eye problems.

Information and communication technologies can play a crucial role in improving the community’s eye health situation and service delivery, for example, developing a reminder system for routine eye exams can help to comply with the recommended frequency. Therefore, improving the availability of eye health services for communities and reducing long queues in hospitals are major challenges for health managers and they involve improving infrastructure and equipment and the training of specialized human resources since a lack of these constitutes a risk factor for the accessibility of eye health services.

In this context, digital health (telemedicine and artificial intelligence) can play a promising role in achieving universal coverage of health services, especially in remote regions, by provisioning and monitoring health services from a distance. Therefore, digital health has the potential to reduce the gap caused by a lack of specialized health professionals [41,42]. However, it has its specificities and requires an appropriate operational structure and accessibility to information and communication technologies as well as training of health professionals for its implementation; therefore, for the Mozambican reality, it can be an achievable objective in the medium and long term.

## 5. Conclusions

The use of eye health services has been lower than expected, given the geographic location in which the study was carried out (urban periphery). Despite the fact that 58.3% of the participants had their eye examinations up to date, the distances to the health centers and the relatively better transport conditions in the studied area make this value lower than desirable. The main barriers to access that explain this situation are overcrowding in hospitals, financial difficulties, self-medication, traditional treatment, buying eyeglasses on the street, and fear of treatment, among others. The study shows that barriers limit the service target to 33%. The groups most vulnerable to poor access to eye health services are low-income people, the illiterate, and farmers. It is necessary to develop more specific intervention plans for this group, and more sectors should cooperate in order to achieve the objectives of the 1E1F program.

## Figures and Tables

**Figure 1 ijerph-19-03916-f001:**
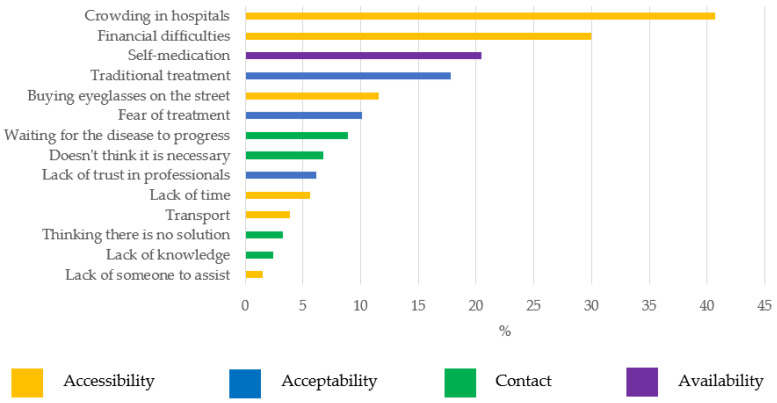
Percentage of mentions of each barrier by respondents. Classification according to the Tanahashi model.

**Figure 2 ijerph-19-03916-f002:**
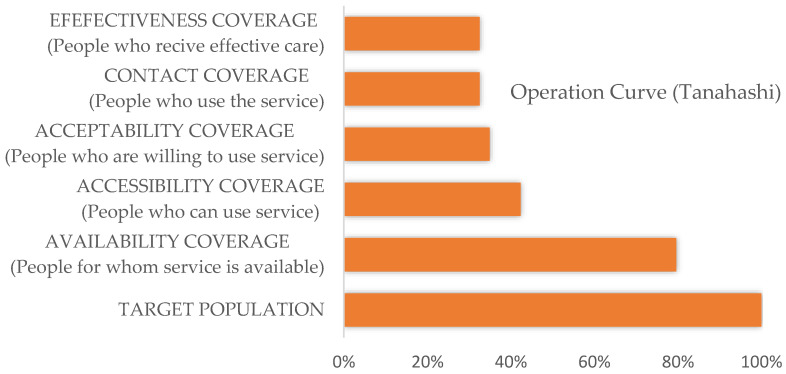
Operation curve according to the Tanahashi model.

**Table 1 ijerph-19-03916-t001:** Description of socio-demographic characteristics and association with the presence of symptoms and the lack of eye examination being up to date.

			With Eye Symptoms			No Eye Examination up to Date		
Variables	N	%	N (%)	OR (95%CI)	aOR (95%CI)	N (%)	OR (95%CI)	aOR (95%CI)
Age (years)								
18–44	202	59.8	97 (48.0)	Ref.	Ref.	94 (46.5)	Ref.	Ref.
45–65	94	27.8	36 (38.3)	0.7 (0.4; 1.1)	n.s.	34 (36.2)	0.7 (0.4; 1.1)	0.6 (0.4; 1.2)
>65	42	12.4	34 (81.0)	4.6 (2.0; 10.4)	n.s.	13 (31.0)	0.5 (0.3; 1.0)	0.4 (0.2; 0.9)
Gender								
Male	183	54.1	88 (48.1)	Ref.	Ref.	70 (38.3)	Ref.	Ref.
Female	155	45.9	79 (51.0)	1.1 (0.7; 1.7)	n.s.	71 (45.8)	1.4 (0.9; 2.1)	n.s.
School level								
Illiterate	50	14.8	30 (60.0)	2.0 (0.9; 4.4)	n.s.	28 (56.0)	11.2 (4.1; 30.9)	n.s.
Primary education	96	28.4	44 (45.8)	1.2 (0.6; 2.2)	n.s.	54 (56.3)	11.4 (4.5; 28.9)	n.s.
Secondary education	133	39.3	68 (51.1)	1.4 (0.8; 2.6)	n.s.	53 (39.8)	5.9 (2.3; 14.6)	n.s.
College and above	59	17.5	25 (42.4)	Ref.	Ref.	6 (10.2)	Ref.	Ref.
Family Income (monthly)								
MZN <5000	124	36.7	79 (63.7)	1.5 (0.6; 3.5)	3.6 (1.3; 9.6)	73 (58.9)	35.8 (4.7; 272.61)	36.8 (4.8; 281.7)
MZN 5000–14,000	78	23.1	34 (43.6)	0.7 (0.3; 1.6)	1.4 (0.5; 3.8)	43 (55.1)	30.7 (4.0; 238;1)	28.3 (3.6; 222.5)
MZN 15,000–24,000	73	21.6	27 (37.0)	0.5 (0.2; 1.2)	0.8 (0.3; 2.1)	20 (27.4)	9.4 (1.2; 74.3)	9.0 (1.1; 71.4)
MZN 25,000–34,000	37	10.9	13 (35.1)	0.5 (0.2; 1.3)	0.6 (0.2; 1.7)	4 (10.8)	3.0 (0.3; 28.8)	2.8 (0.3; 26;5)
MZN 35,000 or more	26	7.7	14 (53.8)	Ref.	Ref.	1 (3.8)	Ref.	Ref.
Occupation								
Student *	46	13.6	26 (56.5)	1.4 (0.7; 2.6)	n.s.	15 (32.6)	0.6 (0.3; 1.2)	n.s.
Teacher *	34	10.1	14 (41.2)	0.7 (0.3; 1.4)	n.s.	10 (29.4)	0.6 (0.3; 1.2)	n.s.
Manual worker *	70	20.7	27 (38.6)	0.6 (0.3; 1.0)	0.5 (0.3; 0.9)	38 (54.3)	1.9 (1.1; 3.2)	n.s.
Farmer *	46	13.6	30 (65.2)	2.1 (1.1; 4.1)	n.s.	24 (52.2)	1.6 (0.8; 3.0)	n.s.
Domestic worker *	35	10.4	14 (40.0)	0.7 (0.3; 1.3)	0.3 (0.1; 0.7)	22 (62.9)	2.6 (1.3; 5.4)	n.s.
Manager *	26	7.7	12 (46.2)	0.9 (0.4; 1.9)	n.s.	1 (3.8)	0.1 (0.1; 0.4)	n.s.
Seller *	57	16.9	24 (42.1)	0.7 (0.4; 1.2)	n.s.	27 (47.4)	1.3 (0.7; 2.3)	n.s.
Retired *	24	7.1	20 (83.3)	5.7 (1.9; 17.0)	7.5 (2.3; 23.9)	4 (16.7)	0.3 (0.1; 0.8)	n.s.
Residence								
Muthita *	119	35.2	53 (44.5)	0.7 (0.5; 1.2)	n.s.	54 (45.4)	1.3 (0.8; 2.0)	n.s.
Piloto *	146	43.2	81 (55.5)	1.5 (1.0; 2.4)	2.0 (1.3; 3.4)	54 (37.0)	0.7 (0.5; 1.1)	n.s.
Nthotta *	73	21.6	167 (49.4)	0.8 (0.5; 1.4)	n.s.	33 (45.2)	1.2 (0.7; 2.0)	n.s.
Total	338	100	167 (49.4)			141 (41.7)		

* The reference group is the rest of the population; CI: confidence interval; OR: odds ratio, aOR: adjusted odds ratio; MZN: metical.

**Table 2 ijerph-19-03916-t002:** Barriers to accessing eye health services and associated factors.

			Reported Barrier			Crowding in Hospitals		
	N	%	N (%)	OR (95%CI)	aOR (95%CI)	N (%)	OR (95%CI)	aOR (95%CI)
TOTAL	338	100	228 (67.5)			137 (40.5)		
Age (years)								
18–44	202	59.8	133 (65.8)	0.7 (0.3; 1.4)	n.s	95 (47.0)	11.5 (3.4; 38.6)	12.9 (3.7; 44.9)
45–65	94	27.8	61 (64.9)	0.3 (0.1; 0.8)	n.s	39 (41.5)	9.2 (2.7; 32.0)	11.0 (3.0; 39.9)
>65	42	12.4	34 (81.0)	Ref.	Ref.	3 (7.1%)	Ref.	Ref.
Gender								
Male	183	54.1	116 (63.4)	Ref.	Ref.	68 (37.2)	Ref.	Ref.
Female	155	45.9	112 (72.3%)	1.5 (0.9; 2.4)	n.s	69 (44.5)	1.4 (0.8; 2.1)	n.s.
School level								
Illiterate	50	14.8	45 (90)	3.1 (1.1; 8.9)	5.4 (1.5; 19.8)	23 (46.0	4.17 (1.7; 10.0)	n.s.
Primary education	96	28.4	78 (81.3)	3.5 (1.3; 9.0)	3.2 (1.7; 8.8)	49 (51.0)	5.1 (2.3; 11.2)	n.s.
Secondary education	133	39.3	86 (64.7)	3.0 (1.2; 7.7)	2.2 (0.9; 5.0)	55 (41.4)	3.5 (1.6; 7.4)	n.s.
College and above	59	17.5	19 (32.2)	Ref.	Ref.	10 (16.9)	Ref.	Ref.
Family Income (monthly)								
MZN <5000	124	36.7	110 (88.7)	2.4 (0.7; 8.7)	5.1 (1.4; 17.9)	65 (52.4)	13.2 (3.0; 58.4)	12.9 (2.8; 58.9)
MZN 5000–14,000	78	23.1	56 (71.8)	3.4 (0.9; 12.4)	2.0 (0.6; 6.6)	40 (51.3)	12.6 (2.8; 57.1)	8.7 (1.9; 40.7)
MZN 15,000–24,000	73	21.6	40 (54.8)	2.5 (0.7; 9.4)	1.2 (0.4; 3.7)	23 (31.5)	5.5 (1.2; 25.4)	4.2 (0.9; 19.8)
MZN 25,000–34,000	37	10.9	13 (35.1)	1.2 (0.3; 5.5)	0.8 (0.3; 2.3)	7 (18.9)	2.8 (0.5; 14.7)	2.2 (0.4; 11.9)
MZN 35,000 or more	26	7.7	9 (34.6)	Ref.	Ref.	2 (7.7)	Ref.	Ref.
Occupation								
Student *	46	13.6	25 (54.3)	0.5 (0.3; 1.0)	n.s	17 (37.0)	0.8 (0.4; 1.6)	n.s.
Teacher *	34	10.1	17 (50.0)	0.4 (0.2; 0.9)	n.s	12 (35.3)	0.8 (0.4; 1.6)	n.s.
Manual worker *	70	20.7	50 (71.4)	1.3 (0.7; 2.5)	n.s	44 (62.9)	3.2 (1.8; 5.5)	2.2 (1.1; 3.4)
Farmer *	46	13.6	43 (93.5)	8.3 (2.5; 27.4)	n.s	17 (37.0)	0.8 (0.4; 1.6)	n.s.
Domestic worker*	35	10.4	27 (77.1)	1.7 (0.8; 3.9)	n.s	20 (57.1)	2.1 (1.0; 4.3)	n.s.
Manager *	26	7.7	10 (38.5)	0.3 (0.1; 0.6)	n.s	2 (7.7)	0.1 (0.0; 0.5)	n.s.
Seller *	57	16.9	41 (71.9)	1.2 (0.7; 2.4)	n.s	25 (43.9)	1.2 (0.7; 2.1)	n.s.
Retired *	24	7.1	15 (62.5)	0.8 (0.3; 1.9)	n.s	0 (0.0)	--	---
Residence								
Muthita *	119	35.2	87 (73.1)	1.5 (0.9; 2.5)	n.s	49 (41.2)	1.0 (0.7; 1.6)	n.s.
Piloto *	146	43.2	91 (62.3)	0.7 (0.4; 1.1)	n.s	55 (37.7)	0.8 (0.5; 1.3)	n.s.
Nthotta *	73	21.6	50 (68.5)	1.1 (0.6; 1.9)	n.s	33 (45.2)	1.3 (0.8; 2.2)	n.s.
			Financial difficulties			Self-medication		
TOTAL	338	100	101 (29.9)			69 (20.4)		
Age (years)								
18–44	202	59.8	42 (20.8)	0.5 (0.2; 1.0)	0.5 (0.2; 1.0)	47 (23.3)	6.1 (1.4; 26.0)	22.2 (2.2; 227.4)
45–65	94	27.8	44 (46.8)	1.6 (0.7; 3.4)	2.2 (0.9; 5.3)	20 (21.3)	5.4 (1.2; 24.3)	19.6 (2.0; 194.0)
>65	42	12.4	15 (35.7)	Ref.	Ref.	2 (4.8)	Ref.	Ref.
Gender								
Male	183	54.1	46 (25.1)	Ref.	Ref.	38 (20.8%)	Ref.	Ref.
Female	155	45.9	55 (35.5)	1.6 (1.0; 2.6)	n.s	31 (20.0)	0.9 (0.6; 1.6)	
School level								
Illiterate	50	14.8	25 (50.0)	8.8 (3.2; 24.3)	n.s	9 (18.0)	2.4 (0.7; 7.6)	n.s
Primary education	96	28.4	40 (41.7)	6.3 (2.4; 16.1)	n.s	27 (28.1)	4.2 (1.5; 11.7)	n.s
Secondary education	133	39.3	30 (22.6)	2.6 (1.0; 6.6)	n.s	28 (21.1)	2.9 (1.1; 7.9)	n.s
College and above	59	17.5	6 (10.2)	Ref.	Ref.	5 (8.5)	Ref.	Ref.
Family Income (monthly)								
MZN <5000	124	36.7	58 (46.8)	7.3 (2.4; 21.7)	12.5 (3.7; 41.8)	27 (21.8)	7.0 (0.9; 53.7)	9.2 (1.1; 77.9)
MZN 5000–14,000	78	23.1	25 (32.1)	3.4 (1.2; 12.2)	6.2 (1.8; 21.3)	24 (30.8)	11.1 (1.4; 86.8)	12.3(1.5; 103.9)
MZN 15,000–24,000	73	21.6	14 (19.2)	2.0 (0.6; 6.4)	1.9 (0.6; 6.7)	11 (15.1)	4.4 (0.5; 63.2)	5.1 (0.6; 44.8)
MZN 25,000–34,000	37	10.9	4 (10.8)	Ref.	Ref.	6 (16.2)	4.8 (0.5; 42.9)	4.5 (0.5; 41.8)
MZN 35,000 or more	26	7.7	0 (0.0)	--	--	1 (3.8)	Ref.	Ref.
Occupation								
Student *	46	13.6	7 (15.2)	0.4 (0.2; 0.9)	n.s.	7 (15.2)	0.7 (0.3; 1.5)	n.s
Teacher *	34	10.1	10 (29.4)	0.1 (0.4; 2.1)	3.0 (1.1; 8.1)	5 (14.7)	0.6 (0.2; 1.7)	n.s
Manual worker *	70	20.7	21 (30.0)	1.0 (0.6; 1.8)	n.s	18 (25.7)	1.5 (0.8; 2.7)	n.s
Farmer *	46	13.6	26 (56.5)	3.8 (2.0; 7.1)	n.s	6 (13.0)	0.5 (0.2; 1.3)	n.s.
Domestic worker *	35	10.4	10 (28.6)	0.9 (0.4; 2.0)	n.s	10 (28.6)	1.7 (0.8; 3.6)	n.s
Manager *	26	7.7	4 (15.4)	0.4 (0.1; 1.2)	n.s	3 (11.5)	0.5 (0.1; 1.7)	n.s.
Seller *	57	16.9	18 (31.6)	1.1 (0.6; 2.0)	n.s	17 (29.8)	1.9 (1.0; 3.6)	n.s.
Retired *	24	7.1	5 (20.8)	0.6 (0.2; 1.6)	n.s	3 (12.5)	0.5 (0.2; 1.9)	n.s.
Residence								
Muthita *	119	35.2	38 (31.9)	1.2 (0.7; 1.9)	n.s	23 (19.3)	0.9 (0.5; 1.6)	n.s.
Piloto *	146	43.2	40 (27.4)	0.8 (0.5; 1.3)	n.s	32 (21.9)	1.2 (0.7; 2.0)	n.s.
Nthotta *	73	21.6	23 (31.5)	1.1 (0.6; 1.9)	n.s	14 (19.2)	0.9 (0.5; 1.7)	n.s.

***** The reference group is the rest of the population; CI: confidence interval; OR: odds ratio; aOR: adjusted odds ratio; MZN: metical.

**Table 3 ijerph-19-03916-t003:** Barriers to accessing eye health services and associated factors (continuation).

			Traditional Treatment			Buying Eyeglasses on the Street		
	N	%	N (%)	OR (95%CI)	aOR (95%CI)	N (%)	OR (95%CI)	aOR (95%CI)
TOTAL	338	100	60 (17.8)			39 (11.5)		
Age (years)								
18–44 years	202	59.8	24 (11.9)	0.1 (0.1, 0.3)	n.s.	7 (3.5)	0.1 (0.0; 0.2)	0.1 (0.0; 0.1)
45–65 years	94	27.8	16 (17.0)	0.2 (0.1; 0.5)	n.s.	17 (18.1)	0.4 (0.2; 0.9)	0.4 (0.2; 1.1)
>65 years	42	12.4	20 (47.6)	Ref.	Ref.	15 (35.7)	Ref.	Ref.
Gender								
Male	183	54.1	25 (13.7)	Ref.	Ref.	15 (8.2)	Ref.	Ref.
Female	155	45.9	35 (22.6)	1.8 (1.1; 3.2)	n.s.	24 (15.5)	2.1 (1.0; 4.1)	n.s.
School level								
Illiterate	50	14.8	22 (44.0)	22.4 (4.9; 102.0)	n.s.	16 (32.0)	7.4 (2.9; 18.6)	n.s.
Primary education	96	28.4	18 (18.8)	6.6 (1.5; 29.5)	n.s.	15 (15.6)	2.9 (1.2; 7.1)	n.s.
Secondary education	133	39.3	18 (13.5)	4.5 (1.0; 20.0)	n.s.	8 (6.0)	Ref.	Ref.
College and above	59	17.5	2 (3.4)	Ref.	Ref.	0 (0.0)	--	--
Family Income (monthly)								
MZN <5000	124	36.7	42 (33.9)	6.1 (1.4; 27.3)	10.5 (1.8; 60.2)	26 (21.0)	9.4 (2.1; 41.0)	15.1 (2.6; 87.7)
MZN 5000–14,000	78	23.1	10 (12.8)	1.8 (0.4; 8.6)	3.7 (0.6; 21.9)	11 (14.1)	5.8 (1.2; 27.3)	11.9 (2.0; 69.9)
MZN 15,000–24,000	73	21.6	4 (5.5)	0.7 (0.1; 4.0)	1.4 (0.2; 9.8)	2 (2.7)	Ref.	Ref.
MZN 25,000–34,000	37	10.9	2 (5.4)	0.7 (0.1; 5.2)	1.2 (0.1; 9.9)	0 (0.0)	--	--
MZN 35,000 or more	26	7.7	2 (7.7)	Ref.	Ref.	0 (0.0)	--	--
Occupation								
Student *	46	13.6	3 (6.5)	0.3 (0.1; 1.0)	n.s.	0 (0.0)	--	--
Teacher *	34	10.1	2 (5.9)	0.3 (0.1; 1.1)	n.s.	3 (8.8)	0.7 (0.2; 2.5)	7.3 (1.3; 39.3)
Manual worker *	70	20.7	9 (12.9)	0.6 (0.3; 1.3)	n.s.	7 (10.0)	0.8 (0.3; 1.9)	n.s.
Farmer *	46	13.6	22 (47.8)	6.1 (3.1; 12.0)	3.5 (1.6; 7.5)	16 (34.8)	6.2 (3.0; 13.1)	n.s.
Domestic worker *	35	10.4	5 (14.3)	0.8 (0.3; 2.0)	n.s.	1 (2.9)	0.2 (0.0; 1.5)	n.s.
Manager *	26	7.7	2 (7.7)	0.4 (0.1; 1.6)	n.s.	2 (7.7)	0.6 (0.1; 2.7)	n.s.
Seller *	57	16.9	8 (14.0)	0.7 (0.3; 1.6)	n.s.	6 (10.5)	0.9 (0.4; 2.2)	n.s.
Retired *	24	7.1	9 (37.5)	3.1 (1.3; 7.5)	9.5 (3.2; 28.5)	4 (16.7)	1.6 (0.5; 4.9)	n.s.
Residence								
Muthita *	119	35.2	21 (17.6)	1.0 (0.6; 1.8)	n.s.	18 (15.1)	1.7 (0.9; 3.3)	n.s.
Piloto *	146	43.2	27 (18.5)	1.1 (0.6; 1.9)	n.s.	11 (7.5)	0.5 (0.2; 1.0)	n.s.
Nthotta *	73	21.6	12 (16.4)	0.9 (0.4; 1.8)	n.s.	10 (13.7)	1.3 (0.6; 2.8)	n.s.
			Fear of treatment			Other barriers		
TOTAL	338	100				80 (23.7)		
Age (years)								
18–44 years	202	59.8	7 (3.5)	0.0 (0.0; 0.1)	0.1 (0.0; 0.1)	53 (26.2)	0.7 (0.3; 1.5)	0.7 (0.3; 1.5)
45–65 years	94	27.8	6 (6.4)	0.1 (0.0; 0.2)	0.1 (0.0; 0.2)	13 (13.8)	0.3 (0.1; 0.8)	0.3 (0.1; 0.8)
>65 years	42	12.4	21 (50.0)	Ref.	Ref.	14 (33.3)	Ref.	Ref.
Gender								
Male	183	54.1	15 (8.2)	Ref.	Ref.	44 (24.0)	Ref.	Ref.
Female	155	45.9	19 (12.3)	1.7 (0.8; 3.2)	n.s.	36 (23.2)	1.0 (0.6;1.6)	n.s.
School level								
Illiterate	50	14.8	15 (30.0)	24.9 (3.1; 196.4)	n.s.	13 (26.0)	3.1 (1.1; 8.9)	n.s.
Primary education	96	28.4	11 (11.5)	7.5 (0.9; 59.7)	n.s.	27 (28.1)	3.5 (1.3; 9.0)	n.s.
Secondary education	133	39.3	7 (5.3)	3.2 (0.4; 26.8)	n.s.	34 (25.6)	3.0 (1.2; 7.7)	n.s.
College and above	59	17.5	1 (1.7)	Ref.	Ref.	6 (10.2)	Ref.	Ref.
Family Income (monthly)								
MZN <5000	124	36.7	23 (18.5)	2.7 (0.6; 12.4)	n.s.	30 (24.2)	2.4 (0.7; 8.7)	n.s.
MZN 5000–14,000	78	23.1	7 (9.0)	1.2 (0.2; 6.1)	n.s.	24 (30.8)	3.4 (0.9; 12.4)	n.s.
MZN 15,000–24,000	73	21.6	2 (2.7)	0.3 (0.0; 2.5)	n.s.	18 (24.7)	2.5 (0.7; 9.4)	n.s.
MZN 25,000–34,000	37	10.9	0 (0.0)	--	--	5 (13.5)	1.2 (0.3; 5.5)	--
MZN 35,000 or more	26	7.7	2 (7.7)	Ref.	Ref.	3 (11.5)	Ref.	Ref.
Occupation								
Student *	46	13.6	1 (2.2)	0.2 (0.0; 1.3)	n.s.	8 (17.4)	0.6 (0.3;1.4)	n.s.
Teacher *	34	10.1	1 (2.9)	0.2 (0.0; 1.9)	n.s.	6 (17.6)	0.7 (0.3; 1.7)	n.s.
Manual worker *	70	20.7	1 (1.4)	0.1 (0.0; 0.8)	n.s.	14 (20.0)	0.8 (0.4; 1.5)	n.s.
Farmer *	46	13.6	17 (37.0)	9.5 (4.4; 20.6)	4.9 (1.8; 13.1)	13 (28.3)	1.3 (0.7; 2.7)	n.s.
Domestic worker *	35	10.4	2 (5.7)	0.5 (0.1; 2.2)	n.s.	8 (22.9)	0.9 (0.4; 2.2)	n.s.
Manager *	26	7.7	1 (3.8)	0.3 (0.0; 2.6)	n.s.	4 (15.4)	0.6 (0.2; 1.7)	n.s.
Seller *	57	16.9	3 (5.3)	0.4 (0.1; 1.5)	n.s.	19 (33.3)	1.8 (1.0;3.4)	2.2 (1.1; 4.1)
Retired *	24	7.1	8 (33.3)	5.5 (2.2; 14.2)	n.s.	8 (33.3)	1.7 (0.7; 4.1)	n.s.
Residence								
Muthita *	119	35.2	14 (11.8)	1.3 (0.6; 2.7)	n.s.	31 (26.1)	1.2 (0.7; 2.1)	n.s.
Piloto *	146	43.2	13 (8.9)	0.8 (0.4; 1.6)	n.s.	28 (19.2)	0.6 (0.4; 1.1)	n.s.
Nthotta *	73	21.6	7 (9.6)	0.9 (0.4; 2.2)	n.s.	21 (28.8)	1.4 (0.8; 2.5)	n.s.

***** The reference group is the rest of the population; CI: confidence interval; OR: odds ratio; aOR: adjusted odds ratio; MZN: metical.

## Data Availability

Data are available upon request from the corresponding author, for study purposes only.

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
