# Peer review of "Barriers to Accessing Eye Health Services in Suburban Communities in Nampula, Mozambique"

_ijerph, 2022, doi:10.3390/ijerph19073916_

Round 1

Reviewer 1 Report

See attached file

Plus points:

1.Unique intervention-- one student, one family programme. It should be described fully in this or another paper and its possible impact real or expected.

2. Most other studies are on rural populations, this is on a peri-urban population. More comments on similarities and differences

3. Data Saturation methodology; more comments on differences, advantages.

The findings are presented to a traditional manner. It can be rewritten to bring out what is new knowledge  in methods and results, answer the "so what?" question  on findings and recommendations on next steps. 

Author Response

Dear Reviewer
Thank you for all your suggestions, many of your comments have, undoubtedly, contributed considerably to improve the manuscript, especially regarding UHC.

Reviewer 1

Comments to Authors

  1. Line 120: Location of interviews medical facility.

Response: We have included in the manuscript the place where the interviews were conducted, the University of Lurio.

  1. Line 182: Number of interviews 25

Any comments? Any effect on design and result?

The objectives of qualitative and quantitative studies are different. While a qualitative study aims, in this case, to find out the variety of problems or barriers that participants face in accessing eye care, a quantitative study aims to count the frequency with which these barriers occur in the population. In addition, determining the number of interviews needed is different for each type of study. In qualitative studies the n cannot be predetermined and depends on when saturation of information occurs, see references 22 to 24. In our study saturation occurred after interview 25 when the last five interviews did not provide barriers other than those already mentioned as described in our paper. In quantitative studies we can calculate sample size based on variability, precision, type 1 error and type 2 error.

In the present work, the quantitative study was done with a sample size of 338, corresponding to a 95% confidence interval, a precision of 5% and assuming a prevalence of 50% as no previous data are available.

  1. Up to date examination: American Optometric Association and American of Ophthalmology. How did the population acquire or were supposed to acquire the knowledge? Would this have an impact?

Knowledge will undoubtedly have an impact on the frequency of attendance at reviews, but not on barriers.  (Section 4.3. paragraph. 9)

Thank you for the question, we have included this reflection in the discussion section of the manuscript.

  1. Line 123. Ocular symptoms. Is there a recall effect? Comment.

Indeed, a recall system could increase the periodicity of attendance at the vision services offered in the Mozambican public health system. Unfortunately, this does not happen and the recall effect is null.

Thank you for your question, we have included this reflection in the discussion section of the manuscript. We have added to section 4.3.1 this paragraph:

“Information and communication technologies can play a crucial role in improving the community's eye health situation and service delivery, for example, developing a reminder system for routine eye exams can help to comply with the recommended frequency. Therefore, improving the availability of eye health services for communities and reducing long queues in hospitals is a major challenge for health managers, and it involves improving infrastructure, equipment and training of specialized human resources, since a lack of these constitutes a risk factor for the accessibility of eye health services.”

  1. Age and gender studied: Children less than 18 years were excluded. Children and carers are a special subgroup in UHC Comments on reasons for exclusion. More analysis on gender?

Indeed, the objective was not clearly defined. We have rewritten it to emphasise that we included only adults in the communities studied.

The reason for excluding the children's population is that children's eye health is taken care of directly in schools. However, this research team has recently published a study on them in "Children":

Ref: Dulnério B. Sengo; Isaura I. D. B. Dos Santos; Momade F. Faquihe; Hermenegildo B. J. F. Tomo; Alcino M. Muaprato; Sualé Puchar; Guida M. R. J. Lôbo; Inmaculada López-Izquierdo; Pablo Caballero. The Prevalence of Visual Impairment and Refractive Errors among a Youth Population in Mozambique: Evidence of the Need for Intervention. Children 2021, Volume 8, Issue 10, 892, doi.org/10.3390/children8100892

  1. Differences between ORs and aORs. any comments/possible explanations?

The difference is the method used for its calculation, bivariate or multivariate. However, we have emphasised in the manuscript the differences between an OR and an aOR, highlighting that the nature of the association described by an aOR is in the presence of all the covariates studied and not bivariate.

  1. Analysis of barriers: In view of current strategy of Universal Health Coverage(UHC) and Integrated Person- centred Eye Care (IPEC), the analysis and presentation could be improved to match barrier to strategy for overcoming barrier

Based on source of barrier, eg

 Barriers related to services and their providers; eye health, -- overcrowding, poor surgical results, general health--pharmacists, private sector, (formal -optometrists and informal street vendors) traditional sector, non-health—transport

  1. Barriers related to Persons/Families
  • Self: Determination of and reconciliation with providers on level of visual need, frequency of eye examination, time of eye examination.
  • Finance: and how patients prioritise expenditure.
  • Knowledge: Seeking of knowledge by persons and provision of knowledge (eye health service providers)

 Thanks for the suggestion, We sincerely believe that this idea has been one of the most important contributors to improve the manuscript. We have classified the barriers obtained according to the Tanahashi model published in the WHO bulletin and calculated the operational curve. In addition, in the discussion section we have commented on this classification and added the findings in the conclusions section.

  1. Age related: Children and the elderly are special subgroups in eye health. Children are not included in the study. The elderly are included and maybe should have a special section.

As mentioned above, children have been the subject of another study by this team.

Regarding the elderly, we have included the discussion section 4.3.1 entitled” Barriers to access to eye health services for the elderly”.

  1. Recommendations can then be linked to the analysis of barriers and or recommended as further studies.

Important questions to answer:

What is new? Findings: So what?

We think that with the numerous changes introduced in the manuscript, we answer these questions. In addition, we have introduced recommendations and future studies.

Plus points:

1.Unique intervention-- one student, one family programme. It should be described fully in this or another paper and its possible impact real or expected.

We have described in the introduction section better. However, references 10 and 11 describe the programme in detail. However, we have extended the description of the 1E1F programme.

  1. Most other studies are on rural populations, this is on a peri-urban population. More comments on similarities and differences.

We have added comments to this respect comparing rural and peri-urban zones.

  1. Data Saturation methodology; more comments on differences, advantages.

As mentioned above, the objectives of qualitative and quantitative studies are different. While a qualitative study aims, in this case, to find out the variety of problems or barriers that participants face in accessing eye care, a quantitative study aims to count the frequency with which these barriers occur in the population. In addition, determining the number of interviews needed is different for each type of study. In qualitative studies the n cannot be predetermined and depends on when saturation of information occurs, see references 22 to 24. In our study saturation occurred after interview 25 when the last five interviews did not provide barriers other than those already mentioned as described in our paper. In quantitative studies we can calculate sample size based on variability, precision, type 1 error and type 2 error.

In the present work, the quantitative study was done with a sample size of 338, corresponding to a 95% confidence interval, a precision of 5% and assuming a prevalence of 50% as no previous data are available.

  1. The findings are presented to a traditional manner. It can be rewritten to bring out what is new knowledge  in methods and results, answer the "so what?" question  on findings and recommendations on next steps. 

We have added numerous changes in agreement with the 3 reviewers and hope that the presentation of the findings has improved.

Reviewer 2 Report

This manuscript is a cross-sectional study that aims to identify barriers to accessing eye health services and associated factors in suburban communities of Nampula. This study is a nice contribution to the already existing literature highlighting factors behind health inequities.

However, some punctual aspects need to be improved:

Introduction:

1-The introduction section highlights the relevance of the subject and provides good references to previous body of work. However, I suggest re-organizing the paragraphs by joining those that have the same idea, as following:

-paragraph 1: line 31 to 44: epidemiology and impact of VI.

-paragraph 2: proposed solutions, specifically in Mozambique and the challenges faced (consider moving lines 48-51 to the end of this paragraph to make it more coherent.

-paragraph 3: highlight the novelty of the study (e.g., to date, no previous study assessed etc.), and then the aim of your study.

2- Please replace “current” by “currently” (line 42)

3- Methods:

Methods are valid and detailed.

-Please remove “quick” (line 101): a review must have been thorough.

-Please reformulate the questions in the survey in a more concise way (e.g., “the reason behind” rather than “why they did not seek medical help”, line 127). Same with the other questions “if they knew the recommended frequency”, “why didn't they have their eye examination up to date?” etc.

4- Results:

Results are clear and well categorized.

Please consider reformulating/summarizing the qualitative results of the survey rather than transcribing the answers in paragraph 3.2.1. You can add these details as a supplementary material. Also, please consider writing these results in one coherent and cohesive paragraph.

5- Discussion:

This section discusses well the results from multiple angles and place them into context.

Few suggestions:

- At the beginning of "Discussion", it is a good idea state in the first paragraph, the main information of your work. After that, go ahead to discuss it.

- You could add the lack of healthcare professional as a risk factor and discuss the role of telemedicine and artificial intelligence as potential promising solutions, keeping in mind the limitations in terms of cost and infrastructure.

Title:

11-The title is informative. Please capitalize the first letter of each word.

Author Response

Dear Reviewer
Thank you for all your suggestions, your comments have contributed considerably to improve the manuscript.

Reviewer 2

This manuscript is a cross-sectional study that aims to identify barriers to accessing eye health services and associated factors in suburban communities of Nampula. This study is a nice contribution to the already existing literature highlighting factors behind health inequities.

However, some punctual aspects need to be improved:

Introduction:

1-The introduction section highlights the relevance of the subject and provides good references to previous body of work. However, I suggest re-organizing the paragraphs by joining those that have the same idea, as following:

-paragraph 1: line 31 to 44: epidemiology and impact of VI.

-paragraph 2: proposed solutions, specifically in Mozambique and the challenges faced (consider moving lines 48-51 to the end of this paragraph to make it more coherent.

-paragraph 3: highlight the novelty of the study (e.g., to date, no previous study assessed etc.), and then the aim of your study.

Following your suggestion we have rearranged the paragraphs of the introduction. Thank you very much.

2- Please replace “current” by “currently” (line 42)

Done

3- Methods:

Methods are valid and detailed.

-Please remove “quick” (line 101): a review must have been thorough.

We have misused the adjective “quick”, it is actually a “Rapid review”. As you know a rapid review is a bibliographic search equation, following the PICO equation, but applied only to one or two bibliographic bases.

-Please reformulate the questions in the survey in a more concise way (e.g., “the reason behind” rather than “why they did not seek medical help”, line 127). Same with the other questions “if they knew the recommended frequency”, “why didn't they have their eye examination up to date?” etc.

We have reworded the whole paragraph as suggested

4- Results:

Results are clear and well categorized.

Please consider reformulating/summarizing the qualitative results of the survey rather than transcribing the answers in paragraph 3.2.1. You can add these details as a supplementary material. Also, please consider writing these results in one coherent and cohesive paragraph.

We have followed the recommendations when reporting the results of qualitative studies, although it is true that the study has a qualitative and quantitative part, we believe it is convenient to keep the results of both techniques and not to relegate the results to complementary material.

5- Discussion:

This section discusses well the results from multiple angles and place them into context.

Few suggestions:

- At the beginning of "Discussion", it is a good idea state in the first paragraph, the main information of your work. After that, go ahead to discuss it.

Thanks for the suggestion; we have started the discussion with this first paragraph with the main results.

- You could add the lack of healthcare professional as a risk factor and discuss the role of telemedicine and artificial intelligence as potential promising solutions, keeping in mind the limitations in terms of cost and infrastructure.

The lack of professionals and telemedicine has been included in the discussion. Thank you for these valuable suggestions.

Title:

11-The title is informative. Please capitalize the first letter of each word.

Done

Kind regards

Reviewer 3 Report

This is a wonderful description of barriers to accessing eye health services in suburban communities in Nampula, Mozambique, via the program “1 student, 1 family” (338 participants; individual interviews, clinical eye examination). Despite 7 hospital units and 6 private clinics 41.7% did not have their eye examination up to date. Most cited by respondents were crowding in hospitals (40.7%), financial difficulties (30.0%), self-medication (20.5%), traditional treatment (17, 8%) and buying eyeglasses on the street (11.6%). The groups most vulnerable to poor access to eye health services are low-income people, the illiterate and farmers.

Comments:

Please explain Metical when first mentioned in line 158 (5.000 Meticais about 60 US$ in 2022)

The community of Piloto was at higher risk. Maybe you can add the distance to Nampula City so readers know how far away it is.

Since an eye examination was performed in all persons, it would be of interest to know what eye diseases those people had (serious ones like dense cataract or refractive problems due to aging, etc.). Maybe you can add a Table.

Line 433: .. Mozambique is on the list of countries with a low Human Development 433

Index (0.456), occupying the 181st position in the global ranking. Maybe you might add: … the 181st position (out of 193 states that are members of the UN) in the global ranking.

From my own experience (Namibia, Zambia, Zimbabwe, Nigeria), I can very much confirm the results. They are all correct and should be changed quickly for the eye health of these people.

Author Response

Dear reviewer

Thank you very much for your feedback. We believe that the changes made following your suggestions have improved the manuscript.

Reviewer 3

This is a wonderful description of barriers to accessing eye health services in suburban communities in Nampula, Mozambique, via the program “1 student, 1 family” (338 participants; individual interviews, clinical eye examination). Despite 7 hospital units and 6 private clinics 41.7% did not have their eye examination up to date. Most cited by respondents were crowding in hospitals (40.7%), financial difficulties (30.0%), self-medication (20.5%), traditional treatment (17, 8%) and buying eyeglasses on the street (11.6%). The groups most vulnerable to poor access to eye health services are low-income people, the illiterate and farmers.

Comments:

Please explain Metical when first mentioned in line 158 (5.000 Meticais about 60 US$ in 2022)

We have added the currency conversion from meticais to dollars.

The community of Piloto was at higher risk. Maybe you can add the distance to Nampula City so readers know how far away it is.

Indeed, we have found a higher risk in Piloto than in the rest of the communities studied, but we have not found any justification for this fact, and the distance does not justify it either, as it is not greater than in Muthita or Nthotta.

For this reason, we have suggested in the discussion section future studies to try to clarify this fact.

Since an eye examination was performed in all persons, it would be of interest to know what eye diseases those people had (serious ones like dense cataract or refractive problems due to aging, etc.). Maybe you can add a Table.

Indeed, this data is very interesting, but we did not include it because we have a study pending publication on the results of the visual examinations. Initially we thought of including it but the result was too broad and we are awaiting its publication.

Line 433: Mozambique is on the list of countries with a low Human Development 433, Index (0.456), occupying the 181st position in the global ranking. Maybe you might add: … the 181st position (out of 193 states that are members of the UN) in the global ranking.

 We have included this updated information to the manuscript in section 4.3.

From my own experience (Namibia, Zambia, Zimbabwe, Nigeria), I can very much confirm the results. They are all correct and should be changed quickly for the eye health of these people.

Thank you for all your valuable suggestions.

Round 2

Reviewer 1 Report

Review inserted corrections, some are repeated.

Author Response

Reviewer 1: Review inserted corrections, some are repeated.

Dear reviewer
We have reviewed the final text and there appears to be no duplication of information. This may be due to the "track changes" option being activated. However, if you find that there is duplicate information, please let us know by page, paragraph and line. 
Kind regards